# Determination of the Inaccuracies of Calculated EEG Indices

**DOI:** 10.3390/s20195715

**Published:** 2020-10-08

**Authors:** Mariusz Borawski, Konrad Biercewicz, Jarosław Duda

**Affiliations:** Department of Multimedia Systems, Faculty of Computer Science and Information Technology, West Pomeranian University of Technology Szczecin, Żołnierska 49, 71-210 Szczecin, Poland; mborawski@wi.zut.edu.pl (M.B.); j.duda@am.szczecin.pl (J.D.)

**Keywords:** EEG, ICA, standard deviation

## Abstract

The data obtained as a result of an EEG measurement are burdened with inaccuracies related to the measurement process itself and the need to remove recorded disturbances. The article presents an example of how to calculate the Approach-Withdraw Index (EEG-AW) and Memorization Index (MI) indices in such a way that their inaccuracy resulting from the removal of artifacts can be periodically calculated. This inaccuracy is expressed in terms of standard deviation. This allows you to determine the reliability of the obtained conclusions in the context of examining elements in a 2D computer game created in the Unity engine.

## 1. Introduction

The data obtained as a result of the measurement are burdened with inaccuracies related to the measurement process itself and the necessity to remove recorded disturbances (if any). These data can be used for further calculations, the result of which will be affected by this inaccuracy. The determination of the level of uncertainty for the calculation result may have a significant impact on the interpretation of the results. For example, if a parameter determined as a result of the calculation is subject to change over time, it is important to relate it to the level of inaccuracy. These changes may be due to inaccurate data recording.

Where calculations require the integration of data from different sources, the inaccuracy of the calculation result will be due to the inaccuracy of data from these sources. One type of calculation of data from many sources is transformation in vector space. A simple example is the measurement of the position of an object in two-dimensional space and the change of the coordinate system. Such a measurement can be carried out using two measuring devices with different parameters. The measurement coordinate system may be different from the target coordinate system. In this case, the coordinate system is transformed. Let us assume that one device provides very accurate values and another very inaccurate. If we start to rotate the coordinate system, the level of inaccuracies on individual axes will change. They will grow and shrink, and after a 360-degree rotation they will return to their original values.

The fact that the uncertainty values may decrease during the calculation excludes the use of methods to determine the inaccuracies in which the uncertainty should be determined from the formula for the transformation of the coordinate system. As can be seen from the deduction of the following article, coordinate inaccuracies do not transform in the same way as coordinates. This allows them to decrease even though they do not take negative values.

The formula for the transformation of the coordinate system is very widely used. It is not limited only to the rotation, changing the scale of objects. A transformation determines how the size of an object changes in particular relativity theory, transformations are Fourier, Cosine, Wave, etc. The method of determining inaccuracies for each transformation is the same. The article presents an example of how to calculate EEG indices on inaccuracies expressed by the standard deviation. By default, the Independent Component Analysis (ICA) component(s) associated with interfering signals are removed in the index calculation procedure. As a result, interference signals are a source of inaccuracies This is a source of inaccuracy, which is expressed in standard deviation. ICA is also a transformation, so the formula for the transformation of the coordinate system can be used for calculations. A similar calculation procedure can be used in any other case where any transformation is used in the calculation.

The EEG examination consists of the appropriate distribution [1,2,3] of the electrodes on the skin surface of the skull, which registers potential changes or differences in the potential of different parts of the brain and after appropriate amplification create a record in the form of an electroencephalogram. In this way, they reflect all brain processes, including other processes such as sensorimotor processing, which are related to the mental condition of a person. As a result, many neurological diseases (e.g., epilepsy can be recognized by examining EEG signals [4].

The main problem in analyzing EEG data is signal interference by physiological and technical artifacts such as eye movements, blinking, muscle activity, heartbeat, high electrode impedance, linear noise, and interference from electrical devices [4]. Artifacts can be removed by cutting out a fragment of the EEG signal. This solution causes the loss of information about the brain’s work when the artifact occurs. This becomes particularly problematic when only a few eras are available and artifacts such as blinking or movement are too frequent. Moreover, this approach is inappropriate for working with [5]:Continuous EEG and activity not blocked by events [6];Long-range time correlations [7];Real-time Brain to Computer (BCI) interface applications [8,9] for example speed control of Festo Robotino mobile robot using NeuroSky MindWave EEG headset based brain-computer interface [10] whether the BCI system designed for human-computer based control of IoT based robot (IoRT) [11];Online mental health monitoring [12].

Other proposed methods of rejecting artifacts are based on [5]:Regression in the time or frequency domain [13];Removing eye artifacts [12];Introducing new artifacts into an EEG recording [14,15,16] and, as a result, are not suitable for real time use [12].

In addition to the above-mentioned methods, the ICA can be used for decomposition of data [12] and removal of one of the signal components associated with the given artifact [17,18]. This ensures that the EEG signal remains continuous. The disadvantage of this solution is that removing the component also removes part of the useful signal.

Figure 1 shows a fragment of the ICA component related to an ocular artifact. A rectangle marks the area where the artifact occurred. Outside this area, the ICA component represents an EEG signal not associated with an artifact.

Removing a component also removes the useful part of the information. This affects all the analysis results. Therefore, it would be important to determine to what extent the removal of the ICA artifact-related component has an impact on specific indices such as the index of interests/ motivation—Approach-Withdrawal (AW). The value of the AW index is related to the increase of interest, its drop together with the decrease of interest [19], or indicators (e.g., Memorization Index (MI). The increase of the MI value is related to enhanced memorization [20,21].

Some studies suggest that the goal-oriented approach and withdrawal behavior are governed by two basic incentive systems: the avoidance system and the approach system [22]. As a result of nature’s choice, the motivation of Approach-Withdrawal is deeply rooted in our mind, because for survival it is crucial to distinguish between pleasant and rewarding stimuli that we can approach and dangerous ones that we should avoid [23]. A lot of neuroscience research [24,25] has proven that there is a motivation for approaching the nervous circuits.

Watching users engage in fun, Malone [26] was interested in exploring the theory behind intrinsically motivating science or science in which an individual engages without motivation (reward or punishment). It describes the characteristics of environments that make them internally motivating, with individual motivations such as challenge, fantasy, curiosity, and control [27]. These features can be considered theories on how to make science enjoyable [28]. In a more recent study, Przybylski, Rigby and Ryan [29,30] identified two other game-related motivational factors, autonomy, and competence, derived from self-determination theory [31]. Games with motivating experience can be explained by the concept of flow [32], invented by Csikszentmihalyi to describe a state in which an individual experiences a challenge that expands their competence without being too difficult or too easy (engaging in challenges at the appropriate level depending on the player’s skills) and has clear objectives and immediate feedback on progress [33].

These are only examples of indexes that have been calculated, but a similar application can be used for engagement, emotion, arousal, valence, etc. indexes. To this end, a measure should be introduced to determine the degree of distortion (loss) of information related to the removal of the artifact. The measured values should be recalculated so that they determine the degree of distortion of the calculation results.

For EEG data, useful information is signal variability. Long-term slow changes of the signal are insignificant information that is removed during the detrending process. The measure of signal variability can be, among others, standard deviation. It allows measuring the level of signal variability. By giving the standard deviation, for example, for a specific index value, it is possible to determine the effect of a particular factor (for example, a deleted ICA component) on its value. However, the main problem is how to calculate the standard deviation of an index from the standard deviation of the removed component. Arithmetic operations performed on a standard deviation have a specific form, which makes it necessary to apply appropriately modified formulas during calculations. However, this will not significantly change the index calculation procedure.

This article presents a method for determining the degree of information loss associated with the removal of the artifact. Therefore, we can gain credibility from the index.

## 2. Materials and Methods

A possible typical calculation procedure for an EEG index is as follows:Selection of the analyzed signal fragment and data synchronization;Division of signal into trials or epochs;Removal of biological artifacts I;Pre-processing;Removal of biological artifacts II;Determination of Individual Alpha Frequency (IAF) [34];Designation of the alpha, beta, theta bands (depending on the requirements of a particular index) [35];Determination of total GFP field power [36];Calculation of the index;Index standardization.

### 2.1. Selection of the Analyzed Signal Fragment and Data Synchronization

EEG analysis calculations very often use methods with high computational complexity, and many of them are not fully automatic. Therefore, it is necessary to limit the scope of analyzed data to a certain interesting part. If data from other devices or software (eye-tracker, web-tracking, etc.) are used when analyzing the EEG signal. Data must be synchronized with each other by, for example, establishing a common start to the study from which time is counted.

### 2.2. The Division of the Signal into Trials or Epochs

The EEG signal is often related to certain events, fragments of films displayed. In this case, it is convenient to divide it into smaller fragments related to specific events of interest to the researcher (fragments of films). The EEG signal is then divided into uneven sections. For some types of analysis (for example ERP) the signal is divided into fragments of equal length.

### 2.3. Removal of Biological Artifacts I

In addition to signals from the brain, EEG electrodes also record signals from outside the brain, which are called artifacts. Artifacts generated by the body are called biological artifacts. Most often these are artifacts related to eyeball movement, muscle activity, and heart rate. They are treated as interference and removed from the EEG signal. One way to remove them is to locate them and remove the part of the EEG signal where the artifact occurs. Removing a fragment of the EEG signal means that it loses continuity and index values are not calculated for the moments when the artifact occurred.

Many methods require specific pre-treatment to detect the artifact. Therefore, this operation is performed before the actual preprocessing on the copy of the EEG signal. Once the location of the artifacts has been determined, they are removed from the actual EEG signal, which will then be further processed.

### 2.4. Pre-Processing

The pre-processing process involves the pre-processing of the EEG signal, which may include, for example, the removal of interference from the mains, the removal of recorded noise, the removal of slow-moving signal components, the detection of a malfunctioning EEG electrode and the possible reconstruction of its signal, etc.

### 2.5. Removal of Biological Artifacts II

Artifact removal methods for which preprocessing is the same as for the actual EEG signal are performed after its pre-processing. These methods include methods using Independent Component Analysis (ICA). Their most important advantage is the removal of artifacts without removing fragments of the EEG signal. It does not lose continuity, but part of the information not related to the artifacts is also removed.

ICA allows the separation of signals related to the sources they generate. Each EEG electrode receives a signal from all sources in the brain. The signal received from a given electrode is, therefore, the sum of all these signals, but their share is different. ICA decomposes the signal in such a way that instead of electrode-linked signals, signals linked to the sources that generate them are received. However, it has its limitations. Typically used ICA algorithms separate as many source-related signals as there are electrodes.

Deleting artifacts using ICA consists of finding the component that contains the artifact and deleting it in its entirety (entering zeros as a component value). This component is not only related to the signal source that generates the artifact. Due to a large number of sources, it is associated with several sources, so useful information is also removed.

### 2.6. Determination of Individual Alpha Frequency (IAF)

The bands in which alpha, beta, and theta waves occur are different for each person. For this reason, the IAF frequency in relation to which all other frequencies are determined is determined at the beginning. The IAF value is calculated for the part of the signal where the test person was muted. She was shown a black screen and asked to try not to think about anything.

In the example, the frequency components were determined using a continuous waveform, using an analytical Morse waveform to calculate the IAF. Power Spectral Density (PSD) was then determined:(1)$$P_{i,j}={\left|c_{i,j}\right|}^{2},$$
where $c_{i,j}$ is *j*-th frequency component for the *i*-th electrode, $P_{i,j}$ than PWD *j*-frequency component for the *i*-th electrode. The formula used to estimate the IAF was [37]:(2)$${\mathrm{IAF}}_{i}=\frac{\sum_{j=j_{0}}^{j_{1}}f_{j}P_{i,j}}{\sum_{j=j_{0}}^{j_{1}}P_{i,j}},$$
where $j_{0}$ and $j_{1}$ are the frequency components corresponding to the frequencies closest to 7.5 Hz and 12.5 Hz respectively (in the discrete version of the transformer between the available frequencies there is a constant pitch), $f_{j}$ is the frequency of the *j* frequency component and ${\mathrm{IAF}}_{i}$ is the IAF value for the *i* electrode. Ultimately, the IAF value is calculated as an average of ${\mathrm{IAF}}_{i}$ for all electrodes.

### 2.7. Designation of the Alpha, Beta, Theta Bands (Depending on the Requirements of a Particular Index)

Based on the IAF frequency, alpha, beta, and/or theta bands are determined. In further steps of the calculation, based on the knowledge of the bands, exact values of alpha, beta, and/or theta, field strength, etc. can be determined.

In the example, the values of the bands are assumed to be $\langle IAF-2,IAF+2)$ fot the alpha, $\langle IAF+2,IAF+16\rangle$ for beta and $\langle IAF-6,IAF-2)$ for theta.

### 2.8. Determination of Total GFP Field Power

The indexes selected for the article require the total power of the field to be calculated. It is determined by the amplitudes of the signal components throughout the band. For selected indexes, these are the alpha and theta bands.

A mid-pass filter can be used to determine GFP. The filter is used to filter out frequency components outside the frequency range for which GFP counts. You can then count the GFP from the formula [38]:(3)$${\mathrm{GFP}}_{\mathrm{band},E,j}=\frac{1}{N_{E}}\sum_{i\in E}x_{\mathrm{band},i,j}^{2},$$
where ${\mathrm{GFP}}_{\mathrm{band},E,j}$ the GFP value for a given frequency band, $x_{\mathrm{band},i,j}$–*j*-the EEG signal sample for the *i*-the electrodes considered for the selected frequency band, E is the the set of electrodes considered, $N_{E}$ is the number of these electrodes. The calculations take into account specific sets of electrodes (for example, on the left side of the head, on the right side of the head, etc.).

### 2.9. Index Calculation

Based on the GFP alpha, the Approach Withdrawal Index [35], can be confessed, and based on the GFP theory (Memorization Index) [39]. The AW index indicates how much of the stimulus is motivating and how much it is depressing for the subject [19,38]:(4)$${\mathrm{AW}}_{j}={\mathrm{GFP}}_{\alpha ,\mathrm{right},j}-{\mathrm{GFP}}_{\alpha ,\mathrm{left},j},$$
where ${\mathrm{GFP}}_{\alpha ,\mathrm{right},j}$ is GFP *j*-samples for electrodes placed on the right side of the head and the alpha band, and ${\mathrm{GFP}}_{\alpha ,\mathrm{right},j}$ is the GFP *j*-sample for the electrodes on the left side of the head and the alpha band.

The MI index allows to determine how much stimulus is likely to be remembered by the subject:(5)$${\mathrm{MI}}_{j}={\mathrm{GFP}}_{\theta ,\mathrm{left},j},$$
where ${\mathrm{GFP}}_{\theta ,\mathrm{right},j}$ is GFP *j*-of this sample for the electrodes on the right side of the head and theta band.

### 2.10. Index Standardization

In order to eliminate individual differences between the examined index values are normalized. Normalization is based on the signal fragment in which the test person is muted. She/He is presented with a black screen and asked to try not to think about anything.

The z-score method can be used for standardization, where standardization is done using the formula [38]:(6)$${\overset{Z}{I}}_{j}=\frac{I_{j}-\overset{B}{\bar{I}}}{{\overset{B}{\sigma }}_{I}},$$
where ${\overset{Z}{I}}_{j}$ is the *j*-value of the index after normalization, $I_{j}$ is the *j*-value of the index before normalization, and $\overset{B}{\bar{I}}$ i ${\overset{B}{\sigma }}_{I}$ is the mean value and standard deviation of the indexes for the part of the signal in which the test person has muted.

Taking into account the loss of information when removing artifacts requires a change in the index calculation procedure. Bold text indicates which points in the procedure have changed:Selection of the analyzed signal fragment and data synchronization;Division of signal into trials or epochs;Removal of biological artifacts I;Pre-processing;Removal of biological artifacts II;Determination of Individual Alpha Frequency (IAF);Designation of the alpha, beta, theta bands (depending on the requirements of a particular index);**Determination of the average value and covariance matrix of the deleted data for each electrode****Determination of total GFP field power**;**Calculation of the index**;**Index standardization**.

### 2.11. Determination of the Average Value and Covariance Matrix of the Deleted Data for Each Electrode

The determination of the average value and the covariance matrix is based on the removal value of the ICA component(s). To determine them, the opposite is true for the removal of artifacts. The ICA component(s) associated with the artifact shall be left unchanged and the other components shall be zeroed.

Each frequency band will have different average values and covariance matrices, so for each alpha, beta, and theta band, they are determined separately. A mid-pass filter can be used for this purpose. The filter filters out frequency components outside the frequency range of the selected bands.

The average values are determined from the formula:(7)$${\overset{\mathrm{ICA}}{\bar{x}}}_{i}=\frac{1}{N}\sum_{j=0}^{N-1}{\overset{\mathrm{ICA}}{x}}_{i,j},$$
where ${\overset{\mathrm{ICA}}{x}}_{i,j}$ is *j*-sample of *i*-sample of the ICA component, and *N* is the number of samples, and elements of the covariance matrix from the pattern:(8)$$\sigma_{{\overset{\mathrm{ICA}}{x}}_{i,j},{\overset{\mathrm{ICA}}{x}}_{k,j}}=\frac{1}{N}\sum_{j=0}^{N-1}\left({\overset{\mathrm{ICA}}{x}}_{i,j}-{\overset{\mathrm{ICA}}{\bar{x}}}_{i}\right)\left({\overset{\mathrm{ICA}}{x}}_{k,j}-{\overset{\mathrm{ICA}}{\bar{x}}}_{k}\right).$$

Counting according to the above formulae, one average value and one covariance matrix are obtained for each ICA component (i.e., ultimately one electrode). This does not take into account the variability over time of the degree of information loss. To take into account its changes over time, a time frame should be adopted in which the average value and the covariance matrix will be calculated. For each given ${\overset{\mathrm{ICA}}{x}}_{i,j}$ representing a moment of time *j* average and the covariance matrix are calculated for ${\overset{\mathrm{ICA}}{x}}_{i,j}$ with *j* ranging from $j-N_{W}$ to $j+N_{W}$ where $2N_{W}+1$ is the width of the time frame.

The obtained average values and covariations should be recalculated to obtain values for all electrodes. This is done by the transformation from the ICA coordinate system to the electrode coordinate system. ICA is a transformation of coordinates in space where the number of dimensions equals the number of electrodes. Every moment of time that data is recorded is one point in this space (Figure 2).

The ICA algorithm determines a transformation matrix allowing, based on the knowledge of coordinates of points in the electrodes system, to calculate coordinates in the ICA coordinate system. The reverse matrix allows, based on the knowledge of coordinates of points in the ICA coordinate system, to calculate coordinates in the electrode coordinate system.

The calculated mean values and covariations are defined in the ICA coordinate system. It is necessary to convert them into electrode coordinate systems. The coordinates of the points are calculated using the formula:(9)$$x_{i,j}=\sum_{k=0}^{M-1}m_{i,k}{\overset{\mathrm{ICA}}{x}}_{k,j},$$
where *M* is the number of electrodes and $m_{i,k}$ is the element of the transformation matrix from the ICA coordinate system to the electrode coordinate system.

The average component value of the *i*-th electrode is:(10)$${\bar{x}}_{i}=\frac{1}{N}\sum_{j=0}^{N-1}x_{i,j}.$$

By substituting the Formula (Equation 9) will obtain:(11)$${\bar{x}}_{i}=\frac{1}{N}\sum_{j=0}^{N-1}\sum_{k=0}^{M-1}m_{i,k}{\overset{\mathrm{ICA}}{x}}_{k,j},$$
hence:(12)$${\bar{x}}_{i}=\sum_{k=0}^{M-1}m_{i,k}\frac{1}{N}\sum_{j=0}^{N-1}{\overset{\mathrm{ICA}}{x}}_{k,j},$$
that is:(13)$${\bar{x}}_{i}=\sum_{k=0}^{M-1}m_{i,k}{\overset{\mathrm{ICA}}{\bar{x}}}_{k}.$$

The above formula allows to convert the average value from one coordinate system to another.

The covariance of the *i*-th electrode is:(14)$$\sigma_{i,l}=\frac{1}{N}\sum_{j=0}^{N-1}\left(x_{i,j}-{\bar{x}}_{i}\right)\left(x_{l,j}-{\bar{x}}_{l}\right).$$

By substituting the Formulas (Equation 9) and (Equation 13) will obtain:(15)$$\begin{array}{c} \sigma_{i,l}=\frac{1}{N}\sum_{j=0}^{N-1}\left(\sum_{k=0}^{M-1}m_{i,k}{\overset{\mathrm{ICA}}{x}}_{k,j}-\sum_{k=0}^{M-1}m_{i,k}{\overset{\mathrm{ICA}}{\bar{x}}}_{k}\right)\left(\sum_{p=0}^{M-1}m_{l,p}{\overset{\mathrm{ICA}}{x}}_{p,j}-\sum_{p=0}^{M-1}m_{l,p}{\overset{\mathrm{ICA}}{\bar{x}}}_{p}\right), \end{array}$$
hence:(16)$$\begin{array}{c} \sigma_{i,l}=\frac{1}{N}\sum_{j=0}^{N-1}\left[\sum_{k=0}^{M-1}m_{i,k}\left({\overset{\mathrm{ICA}}{x}}_{k,j}-{\overset{\mathrm{ICA}}{\bar{x}}}_{k}\right)\right]\left[\sum_{p=0}^{M-1}m_{l,p}\left({\overset{\mathrm{ICA}}{x}}_{p,j}-{\overset{\mathrm{ICA}}{\bar{x}}}_{p}\right)\right], \end{array}$$
that is:(17)$$\sigma_{i,l}=\sum_{k=0}^{M-1}\sum_{p=0}^{M-1}m_{i,k}m_{l,p}\frac{1}{N}\sum_{j=0}^{N-1}\left({\overset{\mathrm{ICA}}{x}}_{k,j}-{\overset{\mathrm{ICA}}{\bar{x}}}_{k}\right)\left({\overset{\mathrm{ICA}}{x}}_{p,j}-{\overset{\mathrm{ICA}}{\bar{x}}}_{p}\right).$$

By substituting ${\overset{\mathrm{ICA}}{\sigma }}_{x_{k,j},x_{p,j}}$ will get:(18)$$\sigma_{i,l}=\sum_{k=0}^{M-1}\sum_{p=0}^{M-1}m_{i,k}m_{l,p}{\overset{\mathrm{ICA}}{\sigma }}_{x_{k,j},x_{p,j}}.$$

The Formulas (Equation 13) and (Equation 18) allow the transformation of mean values and covariance from the ICA coordinate system to the electrode coordinate system.

### 2.12. Determination of total GFP Field Power

When calculating GFP for signal samples, the Formula (Equation 3). For the GFP values determined, average values and variants determining the degree of information loss should be determined. This requires the determination of the formulae for the average value and the variance when summing up and raising to the second power. Average value at summation:(19)$$\bar{x_{l}+x_{p}}=\frac{1}{N}\sum_{j=0}^{N-1}\left(x_{l,j}+x_{p,j}\right)=\frac{1}{N}\sum_{j=0}^{N-1}x_{l,j}+\frac{1}{N}\sum_{j=0}^{N-1}x_{p,j}={\bar{x}}_{l}+{\bar{x}}_{p}.$$

Covariance at summation:(20)$$\begin{array}{c} \sigma_{x_{l}+x_{p},x_{o}+x_{k}}=\frac{1}{N}\sum_{j=0}^{N-1}\left(x_{l,j}+x_{p,j}-\bar{x_{l}+x_{p}}\right)\left(x_{o,j}+x_{k,j}-\bar{x_{o}+x_{k}}\right)=\\ \frac{1}{N}\sum_{j=0}^{N-1}\left(x_{l,j}-{\bar{x}}_{l}+x_{p,j}-{\bar{x}}_{p}\right)\left(x_{o,j}-{\bar{x}}_{o}+x_{k,j}-{\bar{x}}_{k}\right). \end{array}$$

By multiplying you get:(21)$$\begin{array}{c} \sigma_{x_{l}+x_{p},x_{o}+x_{k}}=\frac{1}{N}\sum_{j=0}^{N-1}\left(x_{l,j}-{\bar{x}}_{l}\right)\left(x_{o,j}-{\bar{x}}_{o}\right)+\left(x_{p,j}-{\bar{x}}_{p}\right)\left(x_{k,j}-{\bar{x}}_{k}\right)\\ +\left(x_{l,j}-{\bar{x}}_{l}\right)\left(x_{k,j}-{\bar{x}}_{k}\right)+\left(x_{p,j}-{\bar{x}}_{p}\right)\left(x_{o,j}-{\bar{x}}_{o}\right). \end{array}$$
hence:(22)$$\sigma_{x_{l}+x_{p},x_{o}+x_{k}}=\sigma_{x_{l},x_{o}}+\sigma_{x_{p},x_{k}}+\sigma_{x_{l},x_{k}}+\sigma_{x_{p},x_{o}}.$$

On the basis of the above formula, a formula for variance may be given when summing up:(23)$$\sigma_{x_{l}+x_{p}}^{2}=\sigma_{x_{l}}^{2}+\sigma_{x_{p}}^{2}+2\sigma_{x_{l},x_{p}}.$$

The above formula can be written differently:(24)$$\sigma_{x_{l}+x_{p}}^{2}=\sigma_{x_{l}}^{2}+\sigma_{x_{p}}^{2}+2r_{x_{l},x_{p}}\sigma_{x_{l}}\sigma_{x_{p}},$$
where $r_{x_{l},x_{p}}$ is correlation between $x_{l}$ and $x_{p}$.

If the correlation takes the value of zero, the Formula (Equation 23) will take the form:(25)$$\sigma_{x_{l}+x_{p}}^{2}=\sigma_{x_{l}}^{2}+\sigma_{x_{p}}^{2}.$$

When the correlation is 1:(26)$$\sigma_{x_{l}+x_{p}}^{2}={\left(\sigma_{x_{l}}+\sigma_{x_{p}}\right)}^{2},$$
and when is −1:(27)$$\sigma_{x_{l}+x_{p}}^{2}={\left(\sigma_{x_{l}}-\sigma_{x_{p}}\right)}^{2}.$$

The formula for the mean value at the square lift can be derived from the covariance. Covariance between $x_{l}$ and $x_{p}$:(28)$$\sigma_{x_{l},x_{p}}=\frac{1}{N}\sum_{j=0}^{N-1}\left(x_{l,j}-{\bar{x}}_{l}\right)\left(x_{p,j}-{\bar{x}}_{p}\right).$$

By multiplying the elements of the above formula you will obtain:(29)$$\sigma_{x_{l},x_{p}}=\frac{1}{N}\sum_{j=0}^{N-1}\left(x_{l,j}x_{p,j}-x_{l,j}{\bar{x}}_{p}-{\bar{x}}_{l}x_{p,j}+{\bar{x}}_{l}{\bar{x}}_{p}\right).$$

By converting the pattern you will get:(30)$$\sigma_{x_{l},x_{p}}=\frac{1}{N}\sum_{j=0}^{N-1}x_{l,j}x_{p,j}-{\bar{x}}_{l}{\bar{x}}_{p}-{\bar{x}}_{l}{\bar{x}}_{p}+\frac{1}{N}\sum_{j=0}^{N-1}{\bar{x}}_{l}{\bar{x}}_{p},$$
hence:(31)$$\sigma_{x_{l},x_{p}}=\bar{x_{l}x_{p}}-2{\bar{x}}_{l}{\bar{x}}_{p}+\frac{N}{N}{\bar{x}}_{l}{\bar{x}}_{p}=\bar{x_{l}x_{p}}-{\bar{x}}_{l}{\bar{x}}_{p}.$$

The above formula results in an average multiplication formula:(32)$$\bar{x_{l}x_{p}}={\bar{x}}_{l}{\bar{x}}_{p}+\sigma_{x_{l},x_{p}}.$$

Therefore, the formula for the average value at the square to the power will be as follows:(33)$$\bar{x_{l}^{2}}={\left({\bar{x}}_{l}\right)}^{2}+\sigma_{x_{l}}^{2}.$$

Once the GFP has been determined, it is sufficient to know the mean value and the variance for further calculations. To square the variance, an approximate formula can be derived. Multiplication of variance:(34)$$\begin{array}{c} \sigma_{x_{l}\cdot x_{p}}^{2}=\frac{1}{N}\sum_{j=0}^{N-1}{\left(x_{l,i}x_{p,j}-\bar{x_{l}x_{p}}\right)}^{2}=\\ \frac{1}{N}\sum_{j=0}^{N-1}x_{l,i}x_{p,j}-\frac{2}{N}\sum_{j=0}^{N-1}x_{l,i}x_{p,j}\bar{x_{l}x_{p}}+\frac{1}{N}\sum_{j=0}^{N-1}{\bar{x_{l}x_{p}}}^{2}=\\ =\bar{x_{l}^{2}x_{p}^{2}}-2{\bar{x_{l}x_{p}}}^{2}+{\bar{x_{l}x_{p}}}^{2}=\bar{x_{l}^{2}x_{p}^{2}}-{\bar{x_{l}x_{p}}}^{2}. \end{array}$$

To square the variance:(35)$$\sigma_{x_{l}^{2}}^{2}=\bar{x_{l}^{4}}-{\bar{x_{l}^{2}}}^{2}.$$

The result obtained is similar to the formula:(36)$${\bar{x}}_{l}^{2}\sigma_{x_{p}}^{2}+{\bar{x}}_{p}^{2}\sigma_{x_{l}}^{2}+\sigma_{x_{p}}^{2}\sigma_{x_{l}}^{2}.$$

For $\sigma_{x_{p}}^{2}$ i $\sigma_{x_{p}}^{2}$, you can insert average values from the Formula (Equation 33):(37)$$\sigma_{x_{l}}^{2}=\bar{x_{l}^{2}}-{\left({\bar{x}}_{l}\right)}^{2}.$$

The Formula (Equation 36) will take the form:(38)$$\begin{array}{c} {\bar{x}}_{l}^{2}\left(\bar{x_{p}^{2}}-{\bar{x}}_{p}^{2}\right)+{\bar{x}}_{p}^{2}\left(\bar{x_{l}^{2}}-{\bar{x}}_{l}^{2}\right)+\left(\bar{x_{p}^{2}}-{\bar{x}}_{p}^{2}\right)\left(\bar{x_{l}^{2}}-{\bar{x}}_{l}^{2}\right)=\\ {\bar{x}}_{l}^{2}\bar{x_{p}^{2}}-{\bar{x}}_{l}^{2}{\bar{x}}_{p}^{2}+{\bar{x}}_{p}^{2}\bar{x_{l}^{2}}-{\bar{x}}_{p}^{2}{\bar{x}}_{l}^{2}+\bar{x_{p}^{2}}\cdot \bar{x_{l}^{2}}-\bar{x_{p}^{2}}{\bar{x}}_{l}^{2}-{\bar{x}}_{p}^{2}\bar{x_{l}^{2}}+{\bar{x}}_{p}^{2}{\bar{x}}_{l}^{2}. \end{array}$$
that is:(39)$${\bar{x}}_{l}^{2}\sigma_{x_{p}}^{2}+{\bar{x}}_{p}^{2}\sigma_{x_{l}}^{2}+\sigma_{x_{p}}^{2}\sigma_{x_{l}}^{2}=\bar{x_{p}^{2}}\cdot \bar{x_{l}^{2}}-{\bar{x}}_{l}^{2}{\bar{x}}_{p}^{2}.$$

To square the variance:(40)$$\sigma_{x_{l}^{2}}^{2}=\bar{x_{l}^{4}}-{\bar{x_{l}^{2}}}^{2}\approx {\bar{x_{l}^{2}}}^{2}-{\bar{x}}_{l}^{4},$$
that is:(41)$$\sigma_{x_{l}^{2}}^{2}\approx 2{\bar{x}}_{l}^{2}\sigma_{x_{l}}^{2}+\sigma_{x_{l}}^{4}.$$

When analyzing the EEG signal, the mean value is removed during the detrending process, so it can be assumed that $2{\bar{x}}_{l}^{2}=$ 0, hence the above formula will take the form of:(42)$$\sigma_{x_{l}^{2}}^{2}\approx \sigma_{x_{l}}^{4}.$$

Table 1 shows a comparison of the actual standard deviations and those calculated from variance based on the Formula (Equation 42). The actual standard deviations were calculated for data squared. The calculated standard deviation was calculated using the Formula (Equation 42) from the variance calculated from the data. The number of data drawn was 1000, and in the case of the sinusoidal, it was 6184. The standard deviations for the randomised data are the average obtained from the draws, the number of which is given in the repetition column. As you can see, the differences between the calculated and actual values are significant, but proportional to the amplitude of changes in the data. This allows the calculated values to be used to compare the degree of information loss in individual parts of the EEG signal.

GFP is the sum of squares. First, the values are power to squared. The mean value is calculated from the Formula (Equation 33) and the variance from the approximation Formula (Equation 41). The next step is to sum up the calculated value squares. The average value is determined from the Formula (Equation 19). In the case of variance, the covariance necessary to calculate it is unknown. The covariance calculated earlier refers to the value in time. Here, on the other hand, covariance of the data on all electrodes is necessary for calculations.

It can be assumed that the variance sought is within the values determined by the Formulas (Equation 25)–(Equation 27). The loss of information will not exceed the values obtained from these formulae. Therefore, the upper limit of variance can be calculated as:(43)$$\sigma_{\max,x_{l}+x_{p}}^{2}=\max\left\{\sigma_{x_{l}}^{2}+\sigma_{x_{p}}^{2},{\left(\sigma_{x_{l}}+\sigma_{x_{p}}\right)}^{2}\right\}.$$

If a lower limit of variance had to be determined, the formula should be used:(44)$$\sigma_{\min,x_{l}+x_{p}}^{2}=\min\left\{\sigma_{x_{l}}^{2}+\sigma_{x_{p}}^{2},{\left(\sigma_{x_{l}}-\sigma_{x_{p}}\right)}^{2}\right\}.$$

When calculating GFP, the mean values and variance are multiplied by the factor $\frac{1}{N_{E}}$. The mean value and the variance should therefore be determined by multiplying it by a factor of zero variance. Average value when multiplied:(45)$$\bar{sx_{l}}=\frac{1}{N}\sum_{j=0}^{N-1}sx_{l,j}=\frac{s}{N}\sum_{j=0}^{N-1}x_{l,j}=s{\bar{x}}_{l}.$$

The value of the variance in multiplication:(46)$$\sigma_{s\cdot x_{l}}^{2}=\frac{1}{N}\sum_{j=0}^{N-1}{\left(sx_{l,j}-\bar{{sx}_{l}}\right)}^{2}=\frac{s^{2}}{N}\sum_{j=0}^{N-1}{\left(x_{l,j}-\bar{x_{l}}\right)}^{2}=s^{2}\sigma_{x_{l}}^{2}.$$

The average loss of information for GFP:(47)$$\overset{{\mathrm{GFP}}_{\mathrm{band},E}}{{\bar{x}}_{j}}=\frac{1}{N_{E}}\sum_{i\in E}\left({\bar{x}}_{\mathrm{band},1,j}^{2}+\sigma_{\mathrm{band},1,j}^{2}\right).$$

Simplified formula for calculating the upper limit of the loss of information variance for GFP:(48)$$\begin{array}{cc} s_{1,j} & =\frac{1}{N_{E}^{2}}\sigma_{\mathrm{band},1,j}^{4}\\ s_{2,j} & =\frac{1}{N_{E}^{2}}\max\left\{s_{1,j}+\sigma_{\mathrm{band},2,j}^{4},{\left(\sqrt{s_{1,j}}+\sigma_{\mathrm{band},2,j}^{2}\right)}^{2}\right\}\\ s_{3,j} & =\frac{1}{N_{E}^{2}}\max\left\{s_{2,j}+\sigma_{\mathrm{band},3,j}^{4},{\left(\sqrt{s_{2,j}}+\sigma_{\mathrm{band},3,j}^{2}\right)}^{2}\right\}\\ \; & \vdots \\ \overset{{\mathrm{GFP}}_{\mathrm{band},E}}{\sigma_{\max,j}^{2}} & =\frac{1}{N_{E}^{2}}\max\left\{s_{N_{E}-1,j}+\sigma_{\mathrm{band},N_{E},j}^{4},{\left(\sqrt{s_{N_{E}-1,j}}+\sigma_{\mathrm{band},N_{E},j}^{2}\right)}^{2}\right\} \end{array}$$

The formula is simplified because a higher value should be chosen at the stage of each summation. Simplification may cause miscalculations when the values of variance oscillate around zero. Therefore, it is better to set a maximum for each summation separately.

The obtained values are approximate and are intended to determine in which parts of the signal is greater and in which parts of the signal is less loss of information. It is therefore not necessary to set a lower limit of variance.

### 2.13. Calculation of the Index

In the case of MI, the calculated GFP is an index, so there is no need for additional conversions of average values and limits of variance. Average loss of information for the MI index:(49)$${\overset{\mathrm{MI}}{\bar{x}}}_{j}=\overset{{\mathrm{GFP}}_{\theta ,\mathrm{left}}}{{\bar{x}}_{j}}.$$

Loss of information variance for the MI index:(50)$${\overset{\mathrm{MI}}{\sigma^{2}}}_{\max,j}={\overset{{\mathrm{GFP}}_{\theta ,\mathrm{left}}}{\sigma^{2}}}_{\max,j}.$$

The AW index requires the difference between the GFP values to be determined. The mean value at subtraction can be determined from the formula:(51)$$\bar{x_{l}-x_{p}}=\frac{1}{N}\sum_{j=0}^{N-1}\left(x_{l,j}-x_{p,j}\right)=\frac{1}{N}\sum_{j=0}^{N-1}x_{l,j}-\frac{1}{N}\sum_{j=0}^{N-1}x_{p,j}={\bar{x}}_{l}-{\bar{x}}_{p}.$$

Substraction:(52)$$\begin{array}{c} \sigma_{x_{l}-x_{p},x_{o}-x_{k}}=\frac{1}{N}\sum_{j=0}^{N-1}\left(x_{l,j}-x_{p,j}-\bar{x_{l}-x_{p}}\right)\left(x_{o,j}-x_{k,j}-\bar{x_{o}-x_{k}}\right)=\\ \frac{1}{N}\sum_{j=0}^{N-1}\left(x_{l,j}-{\bar{x}}_{l}-x_{p,j}+{\bar{x}}_{p}\right)\left(x_{o,j}-{\bar{x}}_{o}-x_{k,j}+{\bar{x}}_{k}\right). \end{array}$$

By multiplying you get:(53)$$\begin{array}{c} \sigma_{x_{l}-x_{p},x_{o}-x_{k}}=\frac{1}{N}\sum_{j=0}^{N-1}\left(x_{l,j}-{\bar{x}}_{l}\right)\left(x_{o,j}-{\bar{x}}_{o}\right)+\left(x_{p,j}-{\bar{x}}_{p}\right)\left(x_{k,j}-{\bar{x}}_{k}\right)\\ -\left(x_{l,j}-{\bar{x}}_{l}\right)\left(x_{k,j}-{\bar{x}}_{k}\right)-\left(x_{p,j}-{\bar{x}}_{p}\right)\left(x_{o,j}-{\bar{x}}_{o}\right). \end{array}$$
hence:(54)$$\sigma_{x_{l}-x_{p},x_{o}-x_{k}}=\sigma_{x_{l},x_{o}}+\sigma_{x_{p},x_{k}}-\sigma_{x_{l},x_{k}}-\sigma_{x_{p},x_{o}}.$$

On the basis of the above formula, a formula for variance may be given for subtraction:(55)$$\sigma_{x_{l}-x_{p}}^{2}=\sigma_{x_{l}}^{2}+\sigma_{x_{p}}^{2}-2\sigma_{x_{l},x_{p}}.$$

The above formula can be written differently:(56)$$\sigma_{x_{l}-x_{p}}^{2}=\sigma_{x_{l}}^{2}+\sigma_{x_{p}}^{2}-2r_{x_{l},x_{p}}\sigma_{x_{l}}\sigma_{x_{p}}.$$

If the correlation is zero, the Formula (Equation 55) will take the form:(57)$$\sigma_{x_{l}-x_{p}}^{2}=\sigma_{x_{l}}^{2}+\sigma_{x_{p}}^{2}.$$

When the correlation is 1, it will be:(58)$$\sigma_{x_{l}-x_{p}}^{2}={\left(\sigma_{x_{l}}-\sigma_{x_{p}}\right)}^{2},$$
and when is −1:(59)$$\sigma_{x_{l}-x_{p}}^{2}={\left(\sigma_{x_{l}}+\sigma_{x_{p}}\right)}^{2}.$$

The upper variance limit for subtraction when calculating the AW index can be calculated from the formula:(60)$$\sigma_{\max,x_{l}-x_{p}}^{2}=\max\left\{\sigma_{x_{l}}^{2}+\sigma_{x_{p}}^{2},{\left(\sigma_{x_{l}}+\sigma_{x_{p}}\right)}^{2}\right\},$$
and the bottom one:(61)$$\sigma_{\min,x_{l}-x_{p}}^{2}=\min\left\{\sigma_{x_{l}}^{2}+\sigma_{x_{p}}^{2},{\left(\sigma_{x_{l}}-\sigma_{x_{p}}\right)}^{2}\right\}.$$

The average loss of information for the AW index:(62)$${\overset{\mathrm{AW}}{\bar{x}}}_{j}=\overset{{\mathrm{GFP}}_{\alpha ,\mathrm{right}}}{{\bar{x}}_{j}}-\overset{{\mathrm{GFP}}_{\alpha ,\mathrm{left}}}{{\bar{x}}_{j}}.$$

The upper limit of the loss of information variance for the AW index:(63)$${\overset{\mathrm{AW}}{\sigma^{2}}}_{\max,j}=\max\left\{\overset{{\mathrm{GFP}}_{\alpha ,\mathrm{right}}}{\sigma_{\max,j}^{2}}+\overset{{\mathrm{GFP}}_{\alpha ,\mathrm{left}}}{\sigma_{\max,j}^{2}},{\left(\overset{{\mathrm{GFP}}_{\alpha ,\mathrm{right}}}{\sigma_{\max,j}}+\overset{{\mathrm{GFP}}_{\alpha ,\mathrm{left}}}{\sigma_{\max,j}}\right)}^{2}\right\}.$$

### 2.14. Index Standardization

Index normalization is to change the position of zero and change the scale. Subtraction and multiplication are used here, where one of the factors has a variance equal to zero. Average value at subtraction:(64)$$\bar{x_{l}-s}=\frac{1}{N}\sum_{j=0}^{N-1}\left(x_{l,j}-s\right)=\frac{1}{N}\sum_{j=0}^{N-1}x_{l,j}-\frac{1}{N}\sum_{j=0}^{N-1}s={\bar{x}}_{l}-s.$$

Substraction:(65)$$\begin{array}{c} \sigma_{x_{l}-s}^{2}=\frac{1}{N}\sum_{j=0}^{N-1}{\left(x_{l,j}-s-\bar{x_{l}-s}\right)}^{2}=\frac{1}{N}\sum_{j=0}^{N-1}{\left(x_{l,j}-s-{\bar{x}}_{l}+s\right)}^{2}=\frac{1}{N}\sum_{j=0}^{N-1}{\left(x_{l,j}-{\bar{x}}_{l}\right)}^{2}=\sigma_{x_{l}}^{2}. \end{array}$$

It follows from the above formula that the value of the variance does not change at subtraction.

The average loss of information for a standardised index:(66)$${\overset{Z}{\bar{I}}}_{j}=\frac{{\bar{I}}_{j}-\overset{B}{\bar{I}}}{{\overset{B}{\sigma }}_{I}},$$
where ${\overset{Z}{\bar{I}}}_{j}$ is the average loss of information for the *j* index after normalisation, ${\bar{I}}_{j}$ is the average loss of information for the *j* index before normalisation.

Loss of information variance for a standardised index:(67)$${\overset{Z}{\sigma^{2}}}_{j}=\frac{{\overset{I}{\sigma^{2}}}_{j}}{{\overset{B}{\sigma^{2}}}_{I}},$$
where ${\overset{Z}{\sigma^{2}}}_{j}$ is a variant of loss of information for the *j*-th index after standardization, ${\overset{I}{\sigma^{2}}}_{j}$ is the variance of loss of information of the *j*-th index before standardization.

### 2.15. Description of Research

For the purpose of illustrating the method developed, a study was conducted on 20 healthy people (4 = female and 16 = male) the average age was 23 years. The person was informed about the course of the examination. Then they signed consent to participate in the study and were seated in a comfortable chair with access to the keyboard and mouse. The next step was to put on the cap and connect the electrodes to the participant’s scalp and connect them to the data recorder of the participant’s brain.

The cap (g.Nautilus Research Wearable EEG Headset) with 24 electrodes placed in AF3, AF4, F3, F4, F7, F8, FC5, FC6, P7, P8, T7, T8, O1, O2, P3, C3, C4, Pz, Fz, Cz, FPz, Fp1 P4, POz and 3 reference electrodes: AFz, FCz, CPz was used. The channels have been distributed according to the 10-10 system, the international EEG electrode distribution system [40]. The electrodes required a dampened socket to improve conductivity. In order to check whether the EEG electrodes are in good contact with the scalp, impedance values were measured with the g.Recorder program. The sampling frequency was 500 Hz.

After the above steps have been taken, a study was started. Before the game (“2D Game Kit” [41], which was downloaded from the Unity Asset Store and adapted for testing in the Unity engine), there was information how to move and the goal to achieve, i.e., three keys had to be collected, which guaranteed the entrance to the room where the last opponent was located, the so called the boss. Then, after clicking the “Play” button, a black screen appeared, lasting 60 s, during which the participant silenced themselves. After that time a picture of the spider popped up (see: Figure 3), which comes from IAPS [42]. This was used to later normalize the EEG signal. The recorded signals in the final stage were used to calculate the EEG indicators. On the basis of the respondent’s involvement and response (in relation to the game), a comparison of responses and indicators was made. This made it possible to what should be improved in order to keep the engagement in the game at a certain level.

Game was recorded at a resolution of 1920 × 1080 using programmed in-game registration. During the game, screenshots were taken at a frequency of 3 shots per seconds. Each screenshot generated a timestamp for the EEG data to determine the start and end position of each section. Screenshots were saved for later use during the data analysis phase. In addition to the EEG, the study used the Eye Tracker (Eyetribe eye-tracker with a frequency of 30 points per second was used to track the eye) to track image elements that were particularly important to the respondent and whether this is likely to be related to a decrease or increase in the engagement index.

### 2.16. Selection of the Analyzed Signal Fragment and Data Synchronization

The whole signal was used in the analyses, omitting only fragments of the signal just after the activation of the EEG cap and just before it was switched off. EEG cap and game event data were used in the research. It was therefore necessary to synchronise the recorded EEG data with the recorded events.

The fieldTrip library was used to synchronize EEG data with events. In turn, the following procedure was used to determine what level of engagement and memory is on a particular screenshot:The time of the screenshot was determined;For the indicated time between −5 s and +5 s, an index of engagement, memory, the standard deviation was determined and where the test person looked (for each EEG and eye-tracker measurement, a timestamp was generated to make it possible) Eyetribe eye-tracker with a frequency of 30 points per second was used to track the eye.

### 2.17. Division of Signal into Trials or Epochs

The division of an EEG signal into epochs or trials is a procedure in which specific time windows are extracted from a continuous EEG signal. Time windows are called “epochs” and are usually closed in time in relation to an event, e.g., to the visual stimulus. Therefore, if we want to isolate epochs from the signal, we should know what segments are of interest for analysis, for example a specific stimulus. In the analyzed example, the following stimuli were chosen to illustrate the method’s operation: the player starting a break in the game, the beginning of a resume after death, the attack by opponents and overcoming a static obstacle.

**Removal of Biological Artifacts I**. Biological artifacts were not removed at this stage. 

### 2.18. Preprocessing

The frequency signal has been removed from the frequency range of $\langle 49\;\mathrm{Hz},51\;\mathrm{Hz}\rangle$ to eliminate interference with the power grid (50 Hz in Europe). In addition, the fixed component of the signal and local trends using the moving average have been removed.

### 2.19. Removal of Biological Artifacts II

ICA components related to ocular artifacts have been identified and removed. The deletion consisted in resetting the first component of ICA. The shape of the course is shown in the Figure 4.

**Determination of Individual Alpha Frequency (IAF)** The IAF was determined, which for the example shown was 9.79 Hz.

**Determination of the Alpha, Beta, Theta Bands** (Depending on the Requirements of a Particular Index). The IAF has determined the alpha bands (7.79 Hz–11.79 Hz) and theta (3.79 Hz–7.79 Hz). 

### 2.20. Determination of the Average Value and Covariance Matrix of the Deleted Data for each Electrode

Average values and covariance for deleted data were determined for the alpha and theta bands. Due to the fact that only the first component has been removed, the average values and covariance matrices for the remaining components are zero. Figure 5 shows the mean values and standard deviations of component 1 for the alpha and theta bands. As a result of filtering the frequency components outside the selected bands, the fixed component was also removed, which resulted in the calculated average values being very close to zero. Standard deviations depend on the variability of the signal in the selected bands. This variability depends on brain activity and recorded interference. The relationship between standard deviation and brain activity makes it a measure of possible loss of information about brain function.

The obtained variance values for the removed component were converted into all electrodes. The removed component represented eye artefacts, hence its share decreases with the distance from the eyes. Figure 6 shows examples of three standard deviation diagrams that have been calculated for electrodes with different eye distances. The closest was the Fpz electrode, and the farthest was P0z.

The calculated average values assumed very low values (about 4% of the standard deviation level. For this reason, and because the average values are insignificant (only specific frequency bands are considered), their calculation is no longer necessary.

### 2.21. Determination of the Total Power of the GFP Field

In the example, a mid-pass filter using a continuous waveform with Morse analysis wavelet was used to determine GFP. The filter was used to filter out frequency components outside the frequency range for which GFP was calculated. For the determined values of the total power of the field, approximate values of the upper limit of variance were calculated, assuming that the mean values are equal to zero (Figure 7).

### 2.22. Calculation of the Index

The indices and their approximate values of the upper variance limit have been calculated. Figure 8 shows a fragment of the AW index and approximate values of the upper boundary of variance.

### 2.23. Index Standardization

The indices and upper limits of variance were normalized and converted into values of standard deviations. The result is shown in Figure 9. Comparing it with the Figure 8 one can see that the standard deviations have changed in proportion to the AW index. The only difference is that they have not shifted along the y axis.

### 2.24. Description of the Four Situations to Apply Standard Deviation

The developed method was used to investigate the player’s interest in a 2D computer game and how the memorization of elements is shaped. Four examples were chosen to show how to use standard deviation in data interpretation: the hero’s transition to the next level, the hero’s rebirth, fighting opponents and overcoming an obstacle. These examples have been chosen to present the compatibility as well as the contradiction of the engagement and memory index with the standard deviation, and whether an artifact occurred at a particular point in time (in the presented analysis, which only serves the purpose of the demonstration, the focus is on eye artefacts only).

### 2.25. Application of Standard Deviation

When analysing the AW and MI indices using standard deviation, it should be returned whether the standard deviation is increasing. If so, it is related to the stimulus for a specific event. In addition, it should be noted whether the standard deviation exceeds the level of the index value many times, so that we know that the removal of the ICA component has strongly influenced the index value. Additionally, it should be analyzed whether an artifact has occurred at the time. As a result, the information obtained leads us to conclude that the conclusions drawn on the basis of the index value should be treated with caution.

## 3. Results

Moving the hero to the next level is a situation in which the player is rewarded for their efforts. (Figure 10a).

The player’s reaction to this event may be important because it will depend on the effort they had to make to overcome the level, which allows for their indirect assessment. Additionally, there is an aspect of curiosity—what will be the next level and what challenges await?

At the time of the event there was a clear increase in AW and a very small increase in MI. The value of standard deviation has increased rapidly. Right after the event, an eye artifact occurred (Figure 10b).

Additionally, among the respondents there were found 5 erroneous, and 15 good interpretations of the AW index.

The hero’s revival is when the character comes to life at a predetermined point. In this situation, when a player was killed in the water, the revival took place in a nearby place (see Figure 11a).

The player’s influence on this event is important because we can determine to what extent the player is motivated to make another attempt to overcome the obstacle. At the time of the death event there was a drop in the AW index (a few hundred seconds before 0) and a very small increase in the MI index. However, the revival event itself caused a very small pattern of the AW index to the previous situation and an increase in the MI index after the hero’s revival. The value of standard deviation has also increased slightly. An eye artifact occurred before the event (see Figure 11b).

In addition, the AW index was found to be 0 erroneous, and so many 3 good interpretations of the AW index, while the MI index was found to be 0 wrong and 3 good.

Fighting monsters is a situation where opponents appear on our path and we have to defeat them to move on (see Figure 12a).

The player’s reaction to this event is important because it will depend on the effort of defeating opponents. As a result, we will be able to roughly estimate whether the rank of opponents is difficult, medium or easy.

Before the event occurs, the AW index records a decrease due to the fact that the player is moving on a platform where nothing is happening, i.e., they just have to squat down to avoid the obstacles of the spikes. When fighting an opponent the AW index slightly increases due to the hero’s escape from being attacked by monsters, and then slightly increases when killing enemies (a few hundredths of a second after escape). The value of standard deviation has increased rapidly. However, no artifacts were detected (see Figure 12b).

In addition, the AW index was found to be 4 erroneous, and so many 10 good interpretations of the AW index, while the MI index was found to be 4 wrong and 10 good.

Overcoming an obstacle, which is to prevent the player from falling into spikes is another challenge that the hero has to overcome in order to move on to the next stage in the game (see Figure 13a).

The player’s reaction to this event depends on the effort put into overcoming the obstacle. As a result, we will be able to determine whether the obstacles are easy, medium or difficult, just like our opponents.

When an event such as jumping to the platform to avoid spikes occurs, the AW index slightly increases, but continues to reach negative values (previously there is a large drop in value, which can be caused by pushing the box forward). After defeating the spikes you can see a slight increase in the AW index as a result of fighting the opponent. In addition, there is a slight increase in the MI index, which may be caused by focusing your eyes on a road sign that shows a ban on drinking water. The standard deviation value for both cases has increased sharply. After the event, an eye artifact occurred (see Figure 13b).

In addition, the AW index was found to be 8 erroneous, and so many 12 good interpretations of the AW index, while the MI index was found to be 4 wrong and 16 good.

## 4. Discussion

Analysing in the first example, the AW index chart when you move to the next level, you can see that its value increases. This is a reaction to the stimulus that starts the break in the game. Given the value of the standard deviation, which is many times higher than the level of the index value, it should be noted that the removal of the ICA component has had a very strong impact on its value. Looking at the course of the removed component you can see that this is mainly related to the elimination of the ocular artifact. Given how much impact it has had on the index, the conclusions drawn from the value of the index should be treated very carefully. The analysed event should be rejected from the research, because there is a good chance of misinterpretation of the player’s reaction. The same should be done for the MI index.

The opposite is true for the hero’s death, where the AW index decreases, but there is a slight increase in the revival. This is a reaction to the stimulus to restart the game. In addition, taking into account the standard deviation, the value of which is insignificant in relation to the index value, it should be noted that the removal of the ICA component has had very little effect on the index value. However, the MI Index has increased minimalistically since the rebirth. This is probably related to remembering an object on the screen. The standard deviation for MI is also at a low level, which indicates that the removal of the ICA component has not significantly affected the index value.

The ideal situation in which no artifact has occurred is visible when fighting monsters. This is evidenced by the low ratio of standard deviation to the AW and MI indices. Observing the values of the AW index we see a decrease, because it is a reaction to the stimulus that is an attack by enemies. When the monsters are killed, you see a slight increase. Low values of the AW index may signal that the opponent is not demanding and that it was not difficult to overcome. Referring to the MI chart you can see that its value is growing significantly. This is a reaction to a stimulus probably related to remembering how the enemy attacks. Taking into account the standard deviation, which does not exceed the index value, it should be taken into account, as in the case of rebirth, that there was no removal of the ocular artifact at that time. As a result, it is not necessary to reject an event from the study.

Referring still to the MI index chart, but for the moment of overcoming an obstacle during which the player focuses their attention on the road sign, you can see a slight increase. Taking into account the standard deviation value, which is many times higher than the level of the index value, it should be noted that the deletion of the ICA component has had a very strong impact on its value, even though it occurred a few seconds after the event. Comparing this with the graph showing the removed ICA component, it can be concluded that this is partly related to the elimination of the ocular artifact. Therefore, the analyzed event for the MI index would have to be discarded from the study as it was the case with the “player’s transition to the next level” event. In the case of the AW index, the standard deviation has much smaller values, but the values of the index itself are very little above zero. From this we can conclude that the player’s response to the stimulus was weak. Due to large values of the standard deviation in relation to the index value, the analyzed event may be qualified for rejection in case of AW index.

The results obtained for the presented situations are almost in line with our expectations, because first of all, moving from easy to next level of interest should be low. We have obtained a high AW index from the study, but due to the high standard deviation the conclusion that interest is high should be rejected. Secondly, the hero’s rebirth should have caused more interest, and this was because the player wanted to overcome the obstacle at all costs. Thirdly, fighting monsters that are defeated by one shot should cause low interest, which is confirmed. Fourthly, for the MI index, we wanted high values when overcoming an obstacle. Although the player was looking at the sign and there was a slight increase, we received a high standard deviation, which indicates, that we should reject the finding.

The presented approach can be used not only for AW or MI indices, but also for other indices, such as index of engagement, emotion, etc. Using the standard deviation we examine the reliability of conclusions. The higher the standard deviation value, the higher the variability of the signal, i.e., lower reliability. However, it should be examined whether the high standard deviation value is not related to the removed artifact.

## Figures and Tables

**Figure 1 sensors-20-05715-f001:**
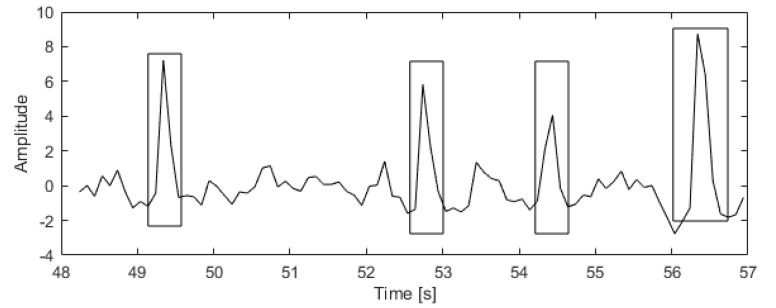
Excerpt from the ICA component related to the eye artifact.

**Figure 2 sensors-20-05715-f002:**
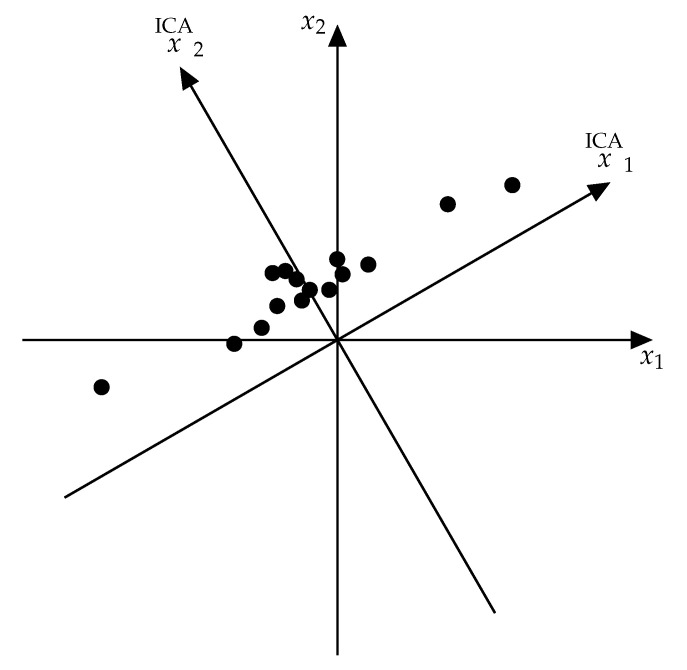
ICA as the transformation of the coordinate system.

**Figure 3 sensors-20-05715-f003:**
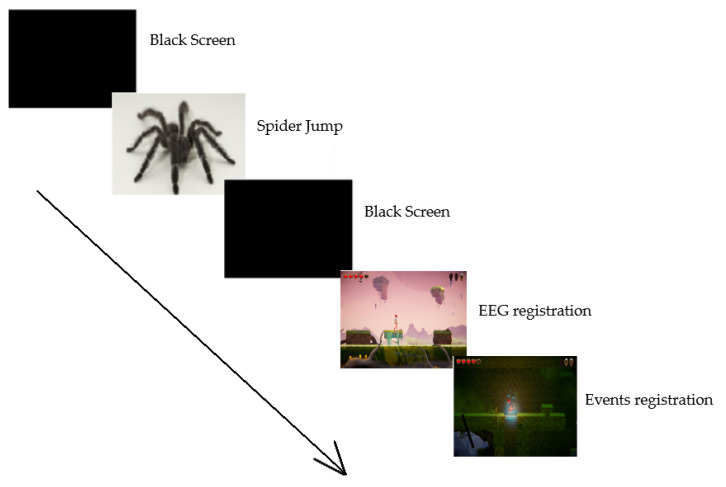
Chronological order of events while playing a computer game.

**Figure 4 sensors-20-05715-f004:**
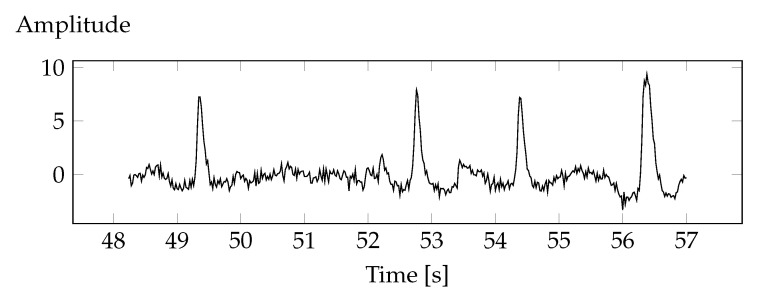
Excerpt from the ICA component containing eye artefacts.

**Figure 5 sensors-20-05715-f005:**
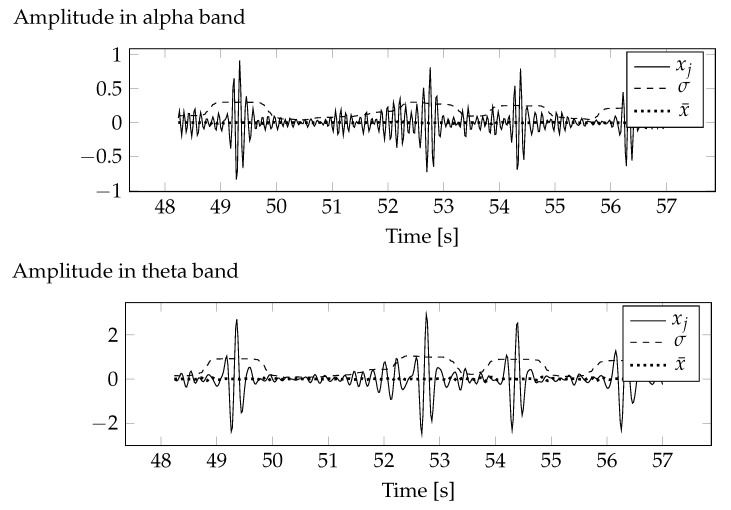
Standard deviations and average values determined for the ICA component to be removed.

**Figure 6 sensors-20-05715-f006:**
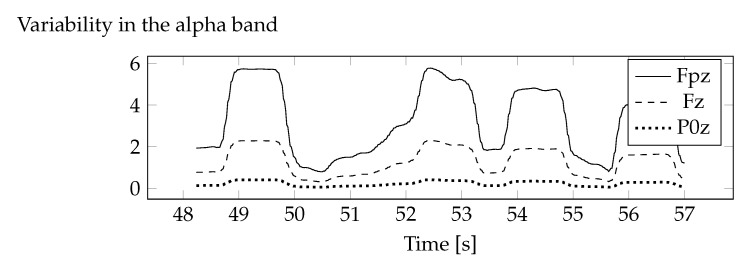
Standard deviations for electrodes at different distances from the eyes.

**Figure 7 sensors-20-05715-f007:**
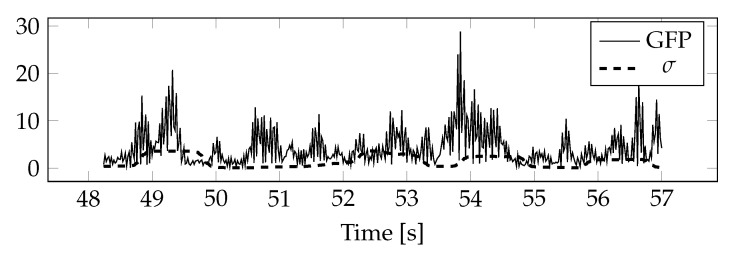
GFP for the alpha band calculated for the electrodes on the right side of the head and its approximate values of the upper limit of standard deviations.

**Figure 8 sensors-20-05715-f008:**
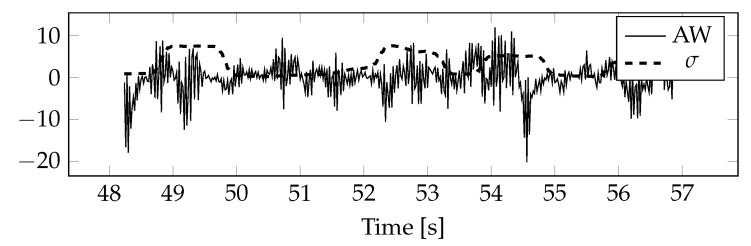
Index AW and its approximate values of the upper limit of standard deviations.

**Figure 9 sensors-20-05715-f009:**
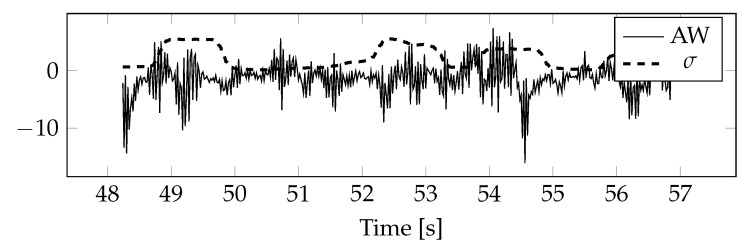
Standardized index AW and its approximate values of the upper limit of standard deviations.

**Figure 10 sensors-20-05715-f010:**
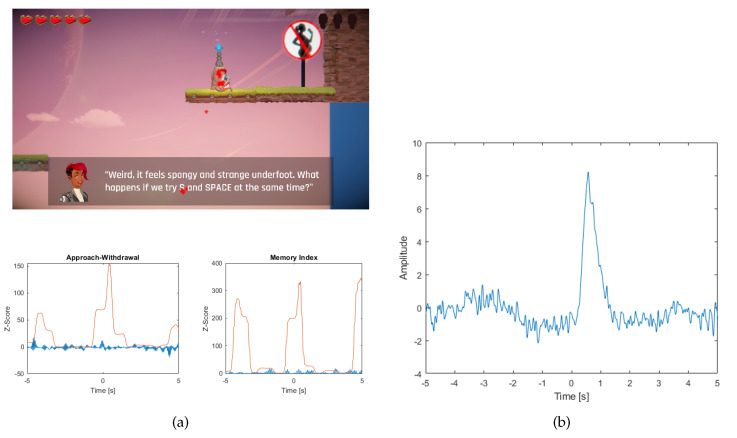
(**a**) The picture shows a screenshot of the game, where the hero is in the middle of the next level. In the picture, the points where the player is looking are marked with red stars. Below are the graphs (blue) of the Approach-Withdrawal and Memory Index and the standard deviation value for both indices (red line) (**b**) A graph showing the occurrence of an ocular artifact for an event related to the hero’s transition to the next level.

**Figure 11 sensors-20-05715-f011:**
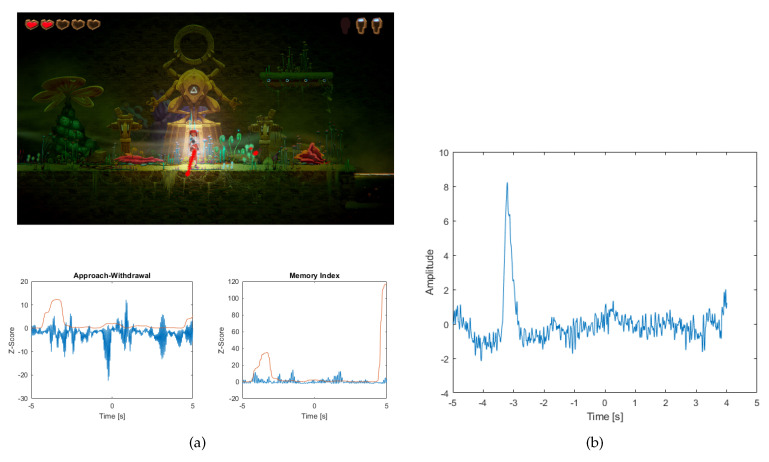
(**a**) The picture shows a screenshot of the game, where the hero revives at a fixed point after death. In the picture, the points where the player is looking are marked with red stars. Below are the graphs (blue) of the Approach-Withdrawal and Memory Index and the standard deviation value for both indices (red line) (**b**) A graph showing the occurrence of an ocular artifact before the hero’s rebirth event.

**Figure 12 sensors-20-05715-f012:**
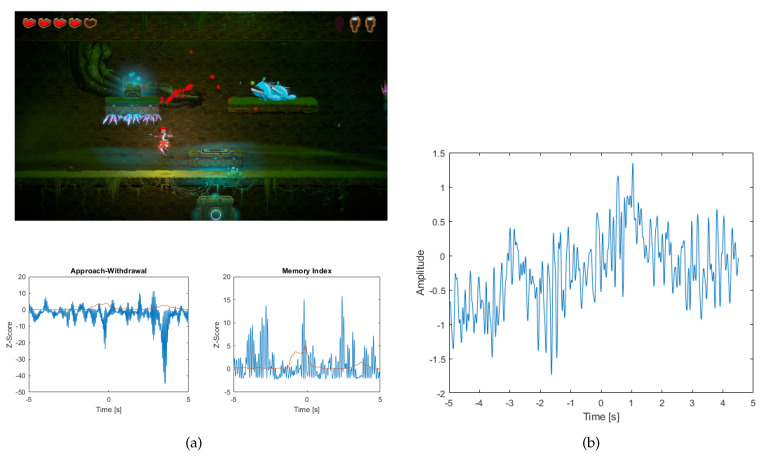
(**a**) The picture shows a screenshot of the game, where the player fights the opponent. In the picture, the points where the player is looking are marked with red stars. Below are the graphs (blue) of the Approach-Withdrawal and Memory Index and the standard deviation value for both indices (red line) (**b**) A graph showing the absence of an artifact for an event related to fighting an opponent.

**Figure 13 sensors-20-05715-f013:**
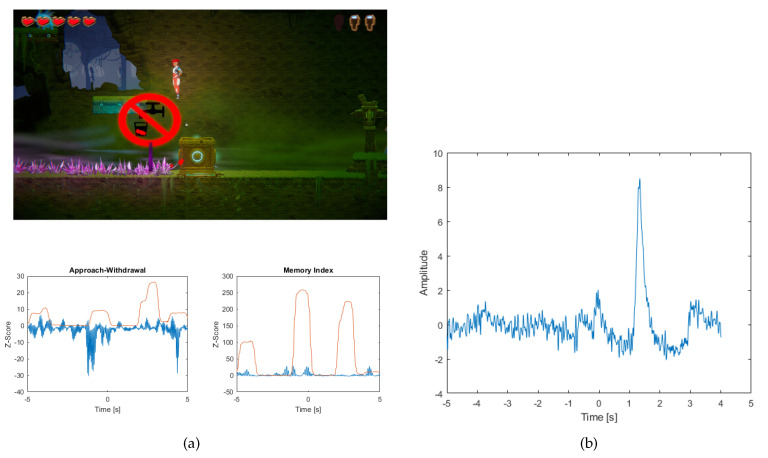
(**a**) The picture shows a screenshot of the game, where the player overcomes an obstacle that is spikes. In the picture, the points where the player is looking are marked with red stars. Below are the graphs (blue) of the Approach-Withdrawal and Memory Index and the standard deviation value for both indices (red line) (**b**) A graph showing the occurrence of an ocular artifact after an obstacle event.

**Table 1 sensors-20-05715-t001:** Comparison of calculated and actual standard deviations.

Type Data	Number Repetitions	σ Calculated	σ Actual
Data drawn with a normal distribution, $\bar{x}=0$, σ=1	1000	1	1.41
Data drawn with a normal distribution, $\bar{x}=0$, σ=9.81	10,000	100.07	141.18
Drawing data of uniform distribution, a=−0.5, b=0.5	1000	0.083	0.074
Drawing data of uniform distribution, a=−5, b=5	10,000	8.33	7.45
Sinusoida (amplitude 1, 20 periods)	1	0.50	0.35
Sinusoida (amplitude 10, 20 periods)	1	50.36	35.43

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
