# Peer review of "Determination of the Inaccuracies of Calculated EEG Indices"

_sensors, 2020, doi:10.3390/s20195715_

Round 1

Reviewer 1 Report

How do you think your findings can be useful applicable to other engineering disciplines?

Introduction is clear, but please explain in more detail in this chapter how the article relates to these topics:

  • Speed control of Festo Robotino mobile robot using NeuroSky MindWave EEG headset based brain-computer interface
  • A Brain–Computer Interface Project Applied in Computer Engineering
  • Electroencephalogram-based brain-computer interface for internet of robotic things
  • Examining the learning efficiency by a brain-computer interface system

Author Response

How do you think your findings can be useful applicable to other engineering disciplines?

This discovery can be used in the BCI field to determine in real time whether a given request for an item in a game/programme is true despite some disruption. Moreover, the method presented in the article is so universal that it can be used for any artifact cut out by the ICA method.

As regards the introduction, BCI-related topics have been taken into account.

Reviewer 2 Report

This paper presents a method to quantify the potential uncertainty in the AW and MI indices that may be associated with the removal of EEG artifacts. They posit that it is important to consider how artifact removal adds uncertainty to EEG-based indices such as AW and MI when interpreting those indices. The added uncertainty represents a loss of information and may lead to inaccurate interpretation of EEG-based indices, which could significantly impact the conclusions of studies using EEG-based indices.

I found the premise of the paper interesting. After multiple readings, however, I am uncertain if I’m understanding the paper as a whole correctly. The organization and presentation of the data collection need to be improved and the EEG analyses to support their premise are not convincing. There are also several grammatical errors.

There are 2 aspects to this paper: the derivation of the method to quantify the standard deviation of the indices that may be attributed to the removed artifact and its utility in interpreting AW and MI in a video game. The derivation is fairly detailed, but the EEG data collection and experiment are not well described. The EEG experiment and indices are presented in the results and discussion, for the most part, rather than appropriately divided into methods, results, and discussion. The authors should reconsider the presentation and organization of the EEG experiment and indices. For example, the “Description of the game” does not describe the actual game, other than state some general features of almost any video game. The descriptions of the 4 video game scenarios used to examine AW and MI and the standard deviation could be explained in the methods. Then in the results would show the plots, figures 11-14 for those 4 scenarios. An explanation of how to use the standard deviation lines to interpret AW and MI would also be useful to describe in the methods before the results. Currently, this is not explained until the beginning of the discussion.  

Lines 38-39 states that “This is a source of inaccuracy, which is expressed in standard deviation” where “this” refers to removing artifactual components from ICA. Line 97, the authors state, “For EEG data, useful information is signal variability,” where signal variability can also be standard deviation, among other metrics. At first, it seems that these ideas oppose each other. In one case the standard deviation of EEG is bad (source of inaccuracy), and in the other case, the standard deviation of EEG is good (useful information). Can the authors better describe the differences in these 2 attributes of standard deviation in EEG?

The method presented addresses “the main problem is how to calculate the standard deviation of an index from the standard deviation of the removed component.” This would address potential information loss, as they suggest. But what about calculating the standard deviation of an index due to the biological components, which would represent potentially meaningful information? If biological components have high standard deviations, is that good or bad, and how would that affect the AW and MI indices’ interpretation? Wouldn’t we lose confidence if the indices' standard deviations are high, regardless of the source?

It is unclear how the method would handle the removal of multiple artifacts for instances when the EEG will have more than just eye artifacts to remove. The only artifact addressed in the manuscript is an ocular artifact.

When interpreting the removed artifact's impact, isn’t it good when removing the clear eye blink had a strong influence on the AW and MI index? Shouldn’t it? The regions to be skeptical of are regions without a clear eye blink that somehow have a strong impact on the indices as evident by the large standard deviations.

The discussion does not put their results into context with the literature. Are the AW and MI results for this video game consistent with what one would expect? If all of the unreliable instances are removed, what would be the interpretation of AW and MI for this game, and how does it compare with other related experiments?

The authors include a comment about social advertising (line 286), which is not described anywhere else.

It is unclear how the data from 2 subjects were used or why it was needed. 

This method's utility would be strengthened with a stronger presentation of mismatches in the interpretation of AW and MI due to artifact removal identified with this method. For example, it would be interesting to quantify the number of potential misinterpretations for each subject while playing the game more than once for the 4 scenarios presented since there is clearly an expected response of AW and MI for each of those scenarios.

There is a lack of details of the EEG equipment and setup. What EEG system was used? How was it set up? Were impedance values measured or some other indicator of good EEG electrode contact? What was the instruction to the subjects? Were they asked to refrain from head movements, clenching of the jaw, blinking, etc.? The map of the sensor’s locations on the headset seems unnecessary (figure 3) as is it a standard headset.

Also, there is a lack of detail about eye tracker. Which eye tracker system was used? Why was it used?

What were the sampling frequencies of all systems? How were they synchronized, as most systems will have different sampling frequencies?

What were the specific stimuli used for creating epochs?

The use of the references in the introduction is sometimes atypical. Ex. Line 42, “[1]-[3] electrodes” doesn’t mean anything to the reader. The main idea of those references needs to be described.

Line 45, “reflect the state of the brain, which is related to the mental condition of a person.” EEG is not only related to the mental condition. It reflects all brain processes, including other processes such as sensorimotor processing. Please revise.

Equation 7, is it missing (1/N) if it is an average?

Figure 6, it seems the second plot is mislabeled? Should it be the theta band?

Author Response

About questions:

  1. The method presented addresses “the main problem is how to calculate the standard deviation of an index from the standard deviation of the removed component.” This would address potential information loss, as they suggest. But what about calculating the standard deviation of an index due to the biological components, which would represent potentially meaningful information? If biological components have high standard deviations, is that good or bad, and how would that affect the AW and MI indices’ interpretation? Wouldn’t we lose confidence if the indices' standard deviations are high, regardless of the source?

Answer:

The article presents the method for calculating the standard deviation of the removed elements. Of course, it is possible to calculate the standard deviation for biological elements, but this is a separate topic, as it should be taken into account, among other things taking into account the fact that not all mathematical actions can be applied due to maintaining correct mathematical rules. Therefore, additional assumptions would be necessary. Nevertheless, we might not be able to carry out the action. That is why we first addressed the topic of standard deviation for the removed elements. In the next phase we plan to develop the method with biological components.

  1. It is unclear how the method would handle the removal of multiple artifacts for instances when the EEG will have more than just eye artifacts to remove. The only artifact addressed in the manuscript is an ocular artifact.

Answer:

The method presented in the article is universal (it can be used for other artifacts) and only an example of how it should be used and how the removed signal affects the result is presented.

  1. When interpreting the removed artifact's impact, isn’t it good when removing the clear eye blink had a strong influence on the AW and MI index? Shouldn’t it? The regions to be skeptical of are regions without a clear eye blink that somehow have a strong impact on the indices as evident by the large standard deviations.

Answer:

A high standard deviation value for the removed artifact is appropriate. An artifact with high variability strongly influences the standard deviation. It's a good point to note that the areas to be skeptical about are those with high standard deviation where there was no artifact. The area in which the artifact occurred should also arouse some scepticism about the interpretation of data. There is no guarantee that the artifact only existed in one ICA component. In the case of an ocular artifact, it happens that some blinks cause artifacts that occur in several ICA components.

The other comments are included in the manuscript.
